# WPI Gel Microstructure and Mechanical Behaviour and Their Influence on the Rate of In Vitro Digestion

**DOI:** 10.3390/foods10051066

**Published:** 2021-05-12

**Authors:** Stephen Homer, Roderick Williams, Allison Williams, Amy Logan

**Affiliations:** CSIRO Agriculture & Food, 671 Sneydes Road, Werribee, VIC 3030, Australia; roderick.williams@csiro.au (R.W.); allison.williams@csiro.au (A.W.); amy.logan@csiro.au (A.L.)

**Keywords:** whey protein, gels, microstructure, particle size, digestion, pH

## Abstract

The influence of microstructure and mechanical properties on the in vitro digestibility of 15% whey protein isolate (WPI) gels was investigated. Gels were prepared via heat set gelation at three pH values (pH 3, 5 and 7), which produced gels with distinct microstructures and mechanical properties. The gels were minced to simulate an oral/chewing phase, which led to the formation particles of various sizes and textures. The minced gels were passed through either an Infogest (pre-set pH of 3) or Glass stomach (dynamic pH) protocol. Gels were digested in the gastric phase for up to 120 min, at which point the extent of digestion was measured by the amount of filterable nitrogen passing through a sieve. The digesta from both gastric methods were passed through an in vitro simulated intestinal phase. A strong link was found between the elasticity of the initial gel and the gel particle size following simulated oral processing, which significantly (*p* < 0.01) affected the rate of digestion in the gastric phase. A weaker correlation was also found between the pH of the gels and the extent of gastric digestion. This work highlights the differences in the rate of gastric digestion, arising from oral processing, which can be attributed to the material properties of the substrate.

## 1. Introduction

Commercial whey protein isolate (WPI) is a highly refined product derived from the bovine serum (whey) proteins of milk. The behaviour of WPI in water is strongly governed by the principle constituent protein β-lactoglobulin (β-lg), which usually comprises more than 50% of the total protein, with the remainder consisting of the other serum proteins found in milk, which includes α-lactalbumin, immunoglobulins, proteose peptones, serum albumin and lactoferrin [1,2]. Whey protein structure and functionality has been reviewed earlier by Guyomarc’h et al. [3] and Nicolai et al. [4]. This behaviour is heavily influenced by electrostatics, with the iso-electric point (IEP) of β-lg generally quoted as pH 5.2 [5]. At pH values close to the IEP, β-lg and WPI solutions form a white suspension of spherical aggregates without the need for heating, which can be redispersed back into solution by adjusting the pH away from the IEP [6,7].

Heating native WPI solutions or suspensions at temperatures above the denaturation temperature of β-lg leads to the formation of irreversible protein aggregates [8,9]. The morphology of the aggregates following heating was shown to be related to that of the suspension/solution prior to heating [6]. When heated at pH values away from the IEP, fine-stranded protein aggregates are formed [10,11]. At pH values close to the IEP, spherical aggregates become irreversibly fixed [12,13]. The types of aggregates have been discussed in detail in the literature [3,14,15].

If heating is performed at concentrations above the critical gelation concentration (C_g_), the aggregates form a percolating network and gels are formed. The value of C_g_ is linked with pH and ionic strength and the trends in behaviour of WPI as a function of pH approximates that of β-lg [16]. Furthermore, the mechanical properties of WPI/β-lg gels vary considerably with pH. At low pH, brittle gels that fracture at low strains are formed while at high pH, gels are more elastic and fracture at higher strains [17]. Currently, there is a need for greater understanding of the links between microstructure, scientific mechanical properties, the sensory/digestive behaviour and the nutritive role these have to play in foods [18].

In terms of nutrition, WPI is considered a complete protein source as it has both a Protein Digestibility-Corrected Amino Acid Score (PDCAAS) and Digestible Indispensable Amino Acid Score (DIASS) score of 1.0 [19,20]. Milk proteins, including WPI, have been used to demonstrate that more solid protein based foods can slow the rate of digestion by reducing the rate of gastric emptying [21,22,23]. Additionally, differences in digestion rate have been shown for various milk proteins, including caseins, WPI and modified caseins [24,25,26]. Floury et al. [27] found differing digestion rates for renneted casein gels versus acid gels with similar composition, illustrating that structure can be used to influence digestion rate. Protein gels formed at various temperatures from a number of protein sources, or having been subjected to processing, also exhibit different gastric digestion rates depending on the heating temperature and the source of the protein [23,28]. Loveday [29] provides a recent and comprehensive review of many of the aspects surrounding protein composition and nutrition.

The difference in digestion rate can have significant physiological effects [30], and the overall impact on satiety may be a combination of digestion rate and hormonal factors. Hence, there is potential that food structures may impact satiety and might be useful in weight management strategies, to assist in the control of drug or nutrient delivery or for modulation of sugar levels in diabetics [31]. A link between digestion rate and satiety has been seen in previous studies [22]. Denatured WPI has been shown to have a greater susceptibility to in vitro gastric hydrolysis than native protein, even when gels are formed during denaturation [23,32]. However, simulated intestinal digestion was found by Singh et al. [23], to be relatively unaffected by food structure, indicating that microstructure may influence digestion and nutrient delivery rates primarily during gastric digestion, thereby impacting the rate of overall digestion. Similar results were also found in vivo for β-lg in milks fed to mini-pigs [21]. A review of in vivo and in vitro digestion highlighted an extensive wealth of literature on the subject but also detailed the need for further research in the area [33].

A number of gastric and intestinal in vitro digestion models exist throughout the world aimed at simulating digestion processes. An international collaboration created the Infogest protocol, which proposed a harmonized method for digesting foods using an in vitro system [34]. This was superseded by Infogest 2.0 [35], and has been applied for the in vitro study of digestion in a large number of laboratories throughout the world. Other in vitro methods have also been developed, such as the Glass stomach method [36]. The key differences between the two methods are the use of a tumbling vial to mix and shear the digesta and a buffer which is pre-set to pH 3 at the start of digestion in the former versus an oscillating probe and a dynamic pH which is controlled down to pH 2 in the latter by slow addition of acid. Comparison of dynamic and static gastric digestion with in vivo digestion has also been performed [37]. It was found that the dynamic model, termed a DIDGI-system, provided a more physiologically accurate process and results were more closely related to in vivo systems.

The aim of this work was to investigate the impact of microstructure and the mechanical properties (elasticity, firmness and breakdown behaviour) on the gastric and intestinal digestion behaviour of WPI gels formed during heat-set gelation at pH 3, 5 and 7. It was hypothesised that firmer/stronger gels may exhibit delayed rates of digestion. As pH is well-known to control gel mechanical firmness and strength [17], it was chosen as the key variable. A secondary objective of the study was to compare the Infogest and Glass stomach protocols for the in vitro examination of simulated gastric digestion. It was anticipated that significant differences in the rates of gastric digestion would be observed between the Infogest and Glass stomach protocols due to either the differences in gel breakdown behaviour or the pH of the gels.

## 2. Materials and Methods

### 2.1. Materials

A commercial WPI powder (WPI 8855, 93.0%) produced using ion exchange and ultra-filtration to ensure nativity, was generously donated by Fonterra, New Zealand. Glutaraldehyde (25%), hexamethyldisilizane (HMDS, Cat #C108, osmium tetroxide (OsO4) and Sorensen’s phosphate buffer (SPB) was purchased from ProSciTech, Kirwan, Australia. Pancreatin (102557 Lot Q6811) was purchased from MP Biochemicals, Wangarra, Australia. OPA (97%), Porcine Pepsin (P6887, lot SLBS5133), bile salt (B3883 lot SLBQ3183V) and all other analytical grade chemicals were purchased from Sigma Aldrich, North Ryde, Australia.

Simulated digestive fluids were prepared according to the method of Minekus et al. [34]; The final compositions were:

Simulated saliva fluid (SSF, pH 7.0): 15.1 mM KCl, 3.7 mM KH_2_PO_4_, 13.6 mM NaHCO_3_, 0.15 mM MgCl_2_ and 0.06 mM (NH_4_)_2_CO_3_

Simulated gastric fluid (SGF, pH 2.0): 6.9 mM KCl, 0.9 mM KH_2_PO_4_, 25 mM NaHCO_3_, 47.2 mM NaCl, 0.12 mM MgCl_2_ and 0.5 mM (NH_4_)_2_CO_3_

In all instances, concentrations presented as percentages are weight by weight percentages.

### 2.2. Methods

#### 2.2.1. Preparation of WPI Gels

The WPI powder was dispersed in milliQ water at a protein concentration of 20% and the dispersion was stored at 4 °C. After ~60 hr, the pH was adjusted to pH 7 by dropwise addition of 1 M NaOH while stirring. This yielded a WPI stock solution, which was split into three equal portions. The pH of two portions were adjusted using orthophosphoric acid (85%) while one was left unadjusted. The pH values of the resulting solutions were 3, 5 and 7. All three were diluted to 15% protein using milliQ water and stored for at least 12 hr at 4 °C. The pH values of the solutions were checked to ensure they remained unchanged and then poured into flexible plastic sausage-like tubes (~20 mm in diameter, ~400–500 mm in length, equating to approximately 150 mL volume) which were sealed at both ends. The WPI sausages were gently tumbled end-over-end right before emersion into a large bucket of gently boiling de-ionised water. The sausages were cooked for 10 min before being removed and placed horizontally in a tray and stored for approximately 12 hr at 4 °C. Preliminary experiments in our laboratory (data not shown) demonstrates that 10 min was sufficient to ensure the formation of homogeneous gels. Replicate samples at each pH were prepared in duplicate, independently from one another using separate preparation solutions. Analysis was performed within 1 day of production.

#### 2.2.2. Compression/Relaxation Testing

The WPI sausages were allowed to warm to room temperature and cut into cylindrical discs with a height of approximately 15 mm. The plastic sausage casing was removed before (for gels prepared at pH 5 and 7) or after (for gels prepared at pH 3) cutting and discarded. Gels prepared at pH 3 were too fragile to be cut without the plastic casing, which helped to reduce fracture or crumbling during cutting before being removed and discarded. The WPI cylinders were individually loaded onto a universal testing machine (Instron 5564, MA, USA) fitted with either a 100 or 500 N load cell, depending on the strength of the sample being tested and determined through preliminary analysis (data not shown) to ensure the limits of each load cell were not exceeded. A probe with a 50 mm diameter was attached to the machine. The load force was zeroed, and the probe was lowered onto the WPI cylinder until a force of 0.1 N was measured. At this point, the height was set to zero and compression commenced at a rate of 0.8 mm.s^−1^ until the sample had been compressed by 7.5 mm. The compression was held for 60 s before the probe was raised at a rate of 0.8 mm.s^−1^ until the zero point. A rest period of 10 s was allowed before the procedure was repeated on the same sample. The force was measured throughout the two compression/relaxation cycles. At least four cylinders were measured per replicate sausage. For each cylinder, the first peak force at maximum compression (A), the force at the end of the first hold period (B) and the peak force during the second compression (C) were obtained and averages (mean values) were calculated for each pH across both replicates (i.e., mean values were calculated from at least 8 measurements).

The peak force at the end of the first and second compressions were designated point A and C respectively, while the force measured at the end of the first relaxation was designated point B. The mean values for points A, B and C were calculated using all replicate measurements.

#### 2.2.3. Scanning Electron Microscopy (SEM)

The WPI gels were allowed to warm to room temperature before being cut into pieces measuring approximately 5 × 2 × 2 mm using a scalpel. The gel pieces were soaked in 2.5% glutaraldehyde in 133 mM Sorensen’s Phosphate Buffer (SPB, pH 7.2). Samples were removed after at least 16 hr, washed with SPB at least 3 times and post-fixed in 1% OsO_4_ in SPB. After 1 hr, the samples were washed with SPB. Dehydration was undertaken by incubations (3 × 15 min per concentration) in a series of ethanol solutions (30, 50, 60, 70, 80, 95 and 100%). Samples were incubated (3 × 10 mins) in HMDS before being air-dried on filter paper in a clean petri dish until all HMDS had evaporated and were then stored in a desiccator.

Prior to imaging, samples were mounted on an aluminium stub with double-sided conductive carbon tape. Samples were then iridium coated using a sputter coater (Cressington 208HRD, Watford, UK). The thickness of the iridium coating was ~6 nm (60 mA for 60 s). Samples were imaged using an electron microscope (Carl Zeiss Merlin Field Emission Scanning, Munich, Germany), operated in the secondary electron (SE) mode at 3 kV.

#### 2.2.4. Protein Content Determination

WPI powder and gel samples were weighed into ceramic boats filled with nickel boat liners. Gels were dried for 1 hr at 102 °C. All samples were stored in a desiccator before analysis. The nitrogen content of each sample was measured in triplicate via nitrogen combustion using a nitrogen determinator (Leco TruMac^®^ N, Baulkham Hills, Australia). The protein content was calculated using a nitrogen conversion factor of 6.38.

#### 2.2.5. Model In-Vitro Digestion Systems

Gels were tested using several in vitro model digestion systems. Three phases of digestion were modelled; the oral, gastric and intestinal. To assess the impact of the gastric method, gastric digestion was performed using two different approaches (Glass stomach and Infogest) on each gel type. Gels were created from independently prepared solutions at each pH enabling digestions to be performed in duplicate for each approach.

##### Simulated Oral Phase

Samples of each gel were minced using a single pass through a domestic mincer (Kenwood MG450) fitted with a 3 mm diameter screen to simulate chewing, according to the recommendations outlined in the Infogest protocol [34]. The minced samples were mixed with SSF at a ratio of 1 g of minced sample for each 1 mL of SSF (typically 10 g and 10 mL respectively). Mixing was conducted for approximately 2 min using either a magnetic stirrer (Infogest protocol) or the reciprocating probe (Glass stomach protocol) before proceeding to the gastric phase. Sub-samples of the minced gels were collected for nitrogen content determination.

##### Simulated Gastric Phase (Infogest Method)

Gastric digestion was simulated following the Infogest protocol [34]. The oral phase mixture of minced gel and SSF was mixed with SGF at a rate of 1 mL of SGF for each 1 g of oral phase mixture (typically the total dilution being 10 g minced sample + 10 mL SSF + 20 mL SGF, including the volume required for the addition of enzyme). The pH was adjusted to pH 3 using 1 M HCl and monitored for approximately 1 min. The sample was warmed to 37 °C and digestion initiated by the addition of porcine Pepsin dissolved in SGF with an activity of 2000 U.mL^−1^ in the total mixture (typically a nett addition of 80,000 units). Samples were incubated for either 30, 60, 90 or 120 min at 37 °C with continuous ‘end over end’ mixing at 20 rpm using a rotary mixer (Ratek RM4, Boronia, Australia) inside an incubator (Jouan EB170, Saint-Herblain, France).

##### Simulated Gastric Phase (Glass Stomach Method)

The dynamic Glass stomach method [36] was investigated as an alternative simulation for in-vitro gastric digestion, consisting of a jacketed glass vessel and a spherical Teflon probe. The internal diameter of the vessel was 40 mm and the internal height was 145 mm. At the bottom of the vessel, a hemisphere design was used to avoid any fluid stagnation zones and to ensure continuous fluid flow. The Teflon probe was 30 mm in diameter and was driven by a stepper motor via a cam, to create a reciprocating movement of the probe where the probe movement relative to the wall of the vessel created a similar flow pattern to that of the contraction waves of the stomach wall [36]. The step motor gain was set to 1 and the time constant for the measurement to 10 s. The Teflon probe was oscillated up and down with an average speed of 10 mm.s^−1^ or 8 s per stroke. Water was re-circulated through the jacket of the Glass stomach to bring the chamber temperature to 37 °C. A sample of minced WPI gel (25 g) was added to 25 mL of SSF in the Glass stomach. This mixture was mixed for 2 min using the oscillations of the Teflon probe before 50 mL of SGF, including an amount of Pepsin to give an overall Pepsin activity of 2000 U mL^−1^, as per the Infogest method. Samples were incubated for either 30, 60, 90 or 120 min. The pH of the mixture was controlled throughout the digestion procedure using a titration manager (TitraLab TIM 854, Radiometer Analytical, Lyon, France) which added 0.05 mL.min^-1^ of 1.2 M HCl with a set point pH of 2.0. In addition, gels prepared at pH 7 were digested using a modified Glass stomach protocol where the pH was adjusted to pH 3 following the addition of the minced gel. This pH was monitored for stability for approximately 1 min, after which point no adjustments were made.

##### Gastric Phase Sample Collection (Infogest and Glass Stomach Methods)

At each time point, ‘undigested’ material was separated from ‘digested’ material by filtration through a sieve with an approximate mesh size of 1.5 mm, and the retentate washed with a small amount of deionised water. The filtrate and washings were weighed, and an aliquot taken to determine the nitrogen content of the filtrate.

A sample of each gastric filtrate (20 g) was mixed with intestinal buffer (11 mL, 1.25 × final concentration pH > 9) raising the pH > 8 to cease enzyme activity and immediately frozen for storage prior to subsequent intestinal digestion. On average, samples were stored for ~30 days. No one sample was stored for longer than 60 days.

##### Simulated Intestinal Phase

Following gastric digestion, the sub-samples of filtrate from each of the gastric digestion time points (30, 60, 90 and 120 min) were carried forward from either the Infogest or Glass stomach method. The following components were added to the gastric filtrates: 1.25 × intestinal buffer; a solution of bile salt; a solution of CaCl_2_ (to 0.3 mM in final mix). The mixture was adjusted to pH 7 and warmed to 37 °C. Intestinal digestion commenced with the addition of Pancreatin yielding a final trypsin activity of 100 U.mL^-1^. The mixture was stirred continuously, and the temperature was kept at 37 °C using a water jacketed vessel on a magnetic stirrer. An aliquot (100 µL) of the full digestion mixture was removed as soon as possible and mixed with 900 µL of 0.1 M NaHCO_3_/0.1% SDS (pH > 9) then heated at 95 °C for 10 min to prevent further digestion. Further aliquots were collected at 2, 5, 10, 15, 20, 30, 45, 60, 90 and 120 min incubation time and treated in a similar manner.

##### Analysis of Digested Material

The effect of oral processing on each gel type was investigated qualitatively by diluting and photographing the minced samples to describe the morphology and relative particle size of the material after the simulated oral phase. The photographs were uploaded to Fiji (Image J.) v. 1.52p [38]. The images were converted to black and white and their contrast and brightness levels were adjusted to aid in visualising the particles present. Image analysis was conducted by following the protocol in the Image J User Guide v. 1.46r [39]. An approximate value of the mean diameter of the particles was calculated from the average particle area by assuming perfect sphericity.

For gastric digestion, the proportion of nitrogen, as a marker for protein, passing through the 1.5 mm sieve relative to the total nitrogen in the freshly minced gel (expressed as a percentage) was determined as an indicator of the extent of gastric digestion. The o-phthaldialdehyde (OPA) method described by Church et al. [40], was adapted for use with a 96 well plate reader and was used to monitor the extent of protein hydrolysis at the end of each gastric time point and throughout intestinal digestion.

The OPA assay buffer was prepared by mixing 0.5 mL 20% SDS (0.1% SDS), 5 mg of OPA (97%, Sigma Aldrich, Australia) in methanol and 5 mg of DTT (DL-Dithiothreitol, Sigma Aldrich, Australia) in water. This solution was made up to 100 mL by the addition of 100 mM sodium tetraborate (Sigma Aldrich, Australia) with a pH of 9.9. Fresh OPA solution was prepared each day of analysis. Samples were diluted in a 0.1 M NaHCO_3_ and 0.1% SDS buffer and analysed in triplicate in a 96 well fluorescence plate with 50 µL diluted sample. The plates were analysed at 37 °C using a microplate reader (Clariostar, BMG Labtech, Ortenberg, Germany). The microplate reader was used to add the OPA solution (200 µL) into the wells at a speed of 300 µL.s^−1^ which was shaken for 300 s (double orbital) at 500 rpm. After 900 s the fluorescence was measured with the following settings: excitation 340 nm, dichroic 393.8 nm and emission 450 nm.

#### 2.2.6. Data Analysis

Results are presented as the mean ± standard error, and one-way analysis of variance (ANOVA) using the data analysis tool within Excel (Microsoft Office Professional Plus, 2013) was used to analyse the data for statistical significance. For measuring differences in the large deformation mechanical behaviour of gels prepared at pH 3, 5 and 7, the compression parameters were used as the dependent factors to determine significance (*p* < 0.05). Similarly, as a function of gastric and intestinal digestion time respectively, differences in filterable nitrogen measured in the gastric filtrate and differences in the normalised OPA reactivity of the intestinal digesta for gels prepare at pH 3, 5 and 7 were compared as a function of time (30, 60, 90 and 120 min) for significance (*p* < 0.01).

## 3. Results

### 3.1. Gel Properties

#### 3.1.1. Gel Appearance and Structure

Heating of the 15% WPI solutions resulted in gel formation for all three pH values (pH 3, 5 and 7). The top row in Figure 1 illustrates the appearance of a representative cross-sectional slice from the gels formed at each pH. The gels prepared at pH 3 and 7 were relatively clear, smooth and semi-transparent, while those prepared at pH 5 were white in appearance and had a much coarser texture that crumbled easily. Representative SEM micrographs for each gel presented in the bottom row of Figure 1 revealed the fine-stranded nature of the aggregates in the gels prepared at pH 3 and 7, as well as the larger spherical aggregates of the gels prepared at pH 5. These observations are consistent with those found throughout the literature [14,15].

#### 3.1.2. Gel Mechanical Properties

Free-standing cylinders of each gel were subjected to compression testing, and representative force versus time traces are shown in Figure 2. The percentage of lost and recovered firmness relative to point A are presented in Table 1.

Gels prepared at pH 7 were firmest (*p* < 0.05) compared to those at pH 5 and 3. The gels prepared at pH 5 and 3 were of similar firmness upon the first compression (A), however differences were measured between gels prepared at pH 5 and pH 3 for the force applied at the end of the first compression hold (B) and for the second compression (C), where gels prepared at pH 3 were least firm. Elasticity can be evaluated by the lost firmness (LF) or recovered firmness (RF) following the initial compression, as outlined in Table 1. Gels prepared at pH 7 were not only firmest, but also the most elastic, having the highest LF and lowest RF values (*p* < 0.05). Gels prepared at pH 5 were less so and lost appreciable amounts of water during the first compression. Gels prepared at pH 3 were brittle and fractured during the first compression. The RF values were in the order pH 7 > pH 5 > pH 3 at which the gels were prepared, and the LF and RF values were in agreement. Images showing the elastic nature, the visible water loss, and the fracturing behaviour of the respective gels are shown in Figure 2. These findings are also consistent with previous studies [17].

### 3.2. In Vitro Digestion

The digestion properties of the three 15% WPI gels were investigated using two digestion procedures, both comprising three simulated phases: oral, gastric and intestinal. The only difference between the two digestion procedures was the use of a Glass stomach [36] or an Infogest [34,35] method for the in vitro gastric digestion model.

#### 3.2.1. Simulated Oral Phase

The simulated oral phase was the first step in the digestion process. Macro photographic black and white images of the minced gels are shown in Figure 3. Gels prepared at pH 3 and 5 were sheared into small particles, the majority of which were smaller than the 1.5 mm sieve mesh size used following gastric digestion, although a small number of larger particles were observed. Conversely, the majority of particles formed during mincing of the gels prepared at pH 7 were larger than the 1.5 mm sieve mesh size. Image analysis yielded mean diameter values of 1.32 ± 0.16, 1.29 ± 0.12 and 4.62 ± 0.75 mm for the minced gels prepared at pH 3, 5 and 7 respectively. These numbers should be considered a guide, however, as they do not indicate the range of sizes and are based on an assumed sphericity, which is not the case. Additionally, the particle sizes relate to the minced gel particles prior to any gastric digestion, while sieving was conducted after gastric digestion at each time point.

Figure 2 and Figure 3 suggest a relationship between the firmness or elasticity of the gel and the extent to which the gel particles were reduced in size via the mincing method employed. Heat-set WPI gels in the high pH region (generally above pH 6) are considerably more elastic than those at low pH [17]. The reasoning for this is ascribed to the existence of intermolecular di-sulphide bonds generated by heat-induced di-sulphide interchange reactions at high pH, whereas these have been found to be absent at low pH [41]. It has also been shown that the width of an extruded material is related to the elasticity of the material, and that elastic materials tend to widen after exiting a die located at the end of the extruder [42]. The process of mincing using a rotating barrel to drive a material through a mincing screen could be considered a similar process to extrusion. Hence, it is reasonable to extrapolate that the large size of the minced particles of gels prepared at pH 7 were linked to their elasticity, especially given that many of them appeared to be substantially larger than the 3 mm screen size. Conversely, the brittle gels prepared at pH 3 fragmented into predominantly smaller particles. Qualitative observation of gels prepared at pH 5 showed that they crumbled easily, despite their relatively firm and elastic nature compared to those prepared at pH 3. This observation and the water loss shown in Figure 2 are related to the relatively large pore structure found in these spherical aggregate gels [43]. These factors led to the relatively small particle sizes shown for this gel following the mincing step. The fact that the same mincing procedure produced very different particle sizes in the different gels does however raise the question of whether the mincing process is representative of in vivo oral processing/chewing in all instances. Indeed, several papers examining the oral breakdown behaviour of whey gels (albeit emulsion gels) and have found similar outcomes in vivo to the observations found in this work [44,45]. This adds some level of confidence that the simulated in vitro oral processing method produced reasonably comparable results to in vivo processes.

#### 3.2.2. Simulated Gastric Digestion (Infogest and Glass Stomach)

##### Filterable Nitrogen

Following the oral processing phase, minced gel/SSF mixtures were fed into the Infogest or Glass stomach in vitro gastric digestion systems, and the results are presented in Figure 4a,b respectively. The extent of gastric digestion was assessed by comparing the amount of nitrogen that passed through a 1.5 mm sieve compared to the total amount of nitrogen in the sample of freshly minced protein gel that was used in the experiment, expressed as a percentage. The mesh size of 1.5 mm was chosen to simulate the pyloric filtering mechanism, which inhibits particles of 0.5–2.0 mm from exiting the stomach into the duodenum [46].

Figure 3 and the image analysis results show that minced gels prepared at pH 3 generally had particles smaller than the 1.5 mm sieve mesh size. However, the minced gels prepared at pH 3 appeared sticky in nature, particularly following simulated gastric digestion using the Infogest method, resulting in some retention of agglomerates larger than 1.5 mm on the sieve at the 30–90 min time points of the Infogest digestion. The stickiness and larger aggregate size of the minced gels prepared at pH 3 was not fully captured by the image analysis due to the dilution step employed prior to taking the photographs and the fact the images were taken prior to simulated gastric digestion. Hence, the mean size of 1.32 mm for these aggregates is an underestimation of the effective sizes which underwent sieving following the Infogest gastric digestion. This was substantiated by measurements of ~80% filterable nitrogen at these time points (Figure 4a). After 120 min of digestion however, the filterable nitrogen level increased to ~100%, indicating a breakdown of the agglomerates to sizes below 1.5 mm at this point. Glass stomach digestion of the gels prepared at pH 3 resulted in nearly 100% filterable nitrogen being obtained at all time points, indicating that adequate size reduction was achieved for almost all of the gel particles to pass through the sieve within the first 30 min (Figure 4b).

Similarly, Figure 3 and the image analysis results show that minced gels prepared at pH 5 had particle sizes much smaller than the 1.5 mm sieve mesh size. They were not sticky like the particles of the minced gels prepared at pH 3. This resulted in all of the minced gel prepared at pH 5 passing through the sieve at all time points over the two-hour time period in both digestion systems; hence, the filterable nitrogen was measured to be around 100% for all time points. (Figure 4a,b). Values over 100% occurred in some instances, and this was traced to evaporation from aliquots used to determine nitrogen content during weighing, creating a small bias in the results towards higher nitrogen levels.

Minced particles of gels prepared at pH 7 prior to gastric digestion are also shown in Figure 3 and were much larger than the sieve mesh size of 1.5 mm. When passed through both gastric digestion systems, less than 20% filterable nitrogen was detected in the filtrate of the minced gels prepared at pH 7 at the 30-min time points (Figure 4a,b). The amount of digested material increased slightly over time to approximately 30% after 2 hr in the Infogest system (Figure 4a). More significant increases in the extent of digestion were seen over time in the Glass stomach system, where filterable nitrogen levels significantly increased from approximately 20% after 30 min to around 45% by the 90-min mark (*p* < 0.01) where levels plateaued (Figure 4b). Taking account of the standard error in measurement, gastric digestion of the minced gels prepared at pH 7 using either method did not yield more than 50% filterable nitrogen, indicating that at least half of the minced gel remained on the sieve throughout the two-hour digestion period.

The key factor affecting whether the gel particles passed through the sieve was their effective size at the filtration stage. However, the effective size of the particles at the filtration stage was governed by two aspects. Firstly, the size of the particles following the mincing step and secondly, the extent of breakdown to smaller particles and the gel stickiness, as in the case of the minced gels prepared at pH 3, following the in vitro gastric digestion phase. Although image analysis yielded similar mean particle sizes for the gels prepared at pH 3 and 5, the stickiness of the gels prepared at pH 3 meant that when sieved (following gastric digestion using the Infogest protocol), the effective particle size was larger than that of the gels prepared at pH 5. It has been found that larger protein particles inhibit the penetration of Pepsin, inhibiting protein hydrolysis in the inner parts of the particles [47]. Given that the size of the minced gel particles prepared at pH 7 prior to gastric digestion were considerably larger than the sieve mesh size, it is unsurprising that a negative relationship can be seen between the gel particle sizes following the oral processing phase and the filterable nitrogen in the filtrate. Overall, the amount of filterable nitrogen was seen to increase in more than half of the samples over the two-hour digestion period, and many others could not increase, as they already yielded 100% values after 30 min. Where increases in filterable nitrogen were seen, the size of the particles would have decreased in these samples either by enzymatic breakdown or via mechanical shear forces (or both) as a function of time. Hence, the extent of simulated gastric digestion, measured by filterable nitrogen levels, was correlated to the size of the gel particles produced during the simulated oral processing phase/mincing step, which in turn was determined by the elasticity of the gel. As such, the elasticity of the gel, albeit indirectly, had a significant effect on the gastric digestibility of the gels used in this study.

##### OPA of Gastric Filtrates

An OPA assay has become a common method used for the quantification of enzymatic protein hydrolysis by Pepsin [26,28,40,48,49]. It was utilised in the current work to analyse the extent of protein hydrolysis for each gel following gastric digestion. The results are presented in Figure 5a,b and have been corrected to take account of washing through the sieve. It is important to understand the relationship between filterable nitrogen and the OPA assay. The OPA assay values are affected by the amount of filterable nitrogen and measures the degree of protein degradation on the molecular level through the amount of α-amino acid groups detected in the filtrate. Filterable nitrogen, on the other hand, is a measure of total solubilised protein.

Gels prepared at pH 3 showed no significant change in OPA reactivity as a function of time, except at the 120-min point during gastric digestion. The increase in OPA reactivity for these gels at the 120-min time point correlates to the increase in filterable nitrogen due to the breakdown of the agglomerated gel particles and is more significant in the Infogest (Figure 4a) than the Glass stomach data (Figure 4b). The gels prepared at pH 3 also yielded the highest OPA reactivity of all three gel types in the early stages of digestion. These two findings suggest that for gels prepared at pH 3, proteolytic activity occurred relatively quickly. The high OPA reactivity readings were due to both the high level of filterable nitrogen and the low pH of the gastric fluid, yielding high enzyme activity and therefore a high number of cleavage sites. Little further hydrolysis occurred after the initial 30 min. The difference in OPA reactivity between the Infogest and Glass stomach digestion filtrates of gels prepared at pH 3 is accounted for by the differences in filterable nitrogen, meaning the extent of protein hydrolysis of gels prepared at pH 3 was unaffected by the gastric digestion method.

Gels prepared at pH 5 passed through the sieve entirely at all time points in both the Infogest and Glass stomach experiments (Figure 4a,b) and hence for these gels, the OPA results were not significantly affected by the levels of filterable nitrogen (*p* < 0.01). The OPA results show that the gels produced at pH 5 initially yielded lower values of OPA reactivity than was seen for gels produced at pH 3, despite the levels of filterable nitrogen being similar (Glass stomach) or higher (Infogest) for pH 5 compared to the pH 3 gels. This indicates a lower extent of protein hydrolysis in the early stages of gastric digestion at higher pH than at lower pH. By the 120-min time point, gels prepared at pH 5 yielded higher OPA reactivity from the Glass stomach method than the Infogest method. These results can be attributed to the increasing Pepsin activity, as the pH was reduced in the Glass stomach by the addition of acid. Kondjoyan et al. [50] detailed measurements of Pepsin activity alongside those published earlier by Pletschke et al. [51], with good agreement between the two. In those results, maximal Pepsin activity occurred at pH 2 and decreased to approximately 30% at a pH of ~3.2 and close to 0% at pH 4–5. By extrapolation between pH 2.0 and 3.2, it’s likely that the activity of Pepsin at pH 3 is around ~40% of its activity at pH 2.

Gels prepared at pH 7 yielded much lower overall OPA results compared to the other gels, which is a consequence of the significantly lower filterable nitrogen (*p* < 0.01), as less protein was available for hydrolysis. As the filterable nitrogen was low throughout the entire experiment, the impact of this on the OPA analysis was also seen throughout the experiment. Data at 90 and 120 min showed a higher OPA reactivity in the filtrate from the Glass stomach compared to the Infogest method, which also correlates well with the filterable nitrogen levels from each gel type.

##### Dynamic Glass Stomach vs. Infogest Gastric Digestion

In addition to the effect of gel elasticity on digestion kinetics, the results presented in Figure 4 and Figure 5 indicate an effect from the method of gastric digestion. Results of gels prepared at pH 3 and 7 show that the Glass stomach method yielded higher values of filterable nitrogen at many time points compared to the Infogest gastric method, although gels prepared at pH 5 yielded 100% filterable nitrogen using both methods. One of the key differences between the two methods is the manner in which shear was imparted on the samples. In the Infogest method, samples were tumbled in a vial, while the Glass stomach method used a probe which had some capacity to impart shear as well as a compressive force, but sometimes resulted in the sample remaining un-mixed at the bottom of the jacketed vessel. Another difference was that the Infogest method set the initial pH of gastric digestion at pH 3, which remained un-monitored during the rest of the experiment, while the dynamic Glass stomach method decreased the pH over time from the initial pH of the combined sample and SGF, through the continuous addition of acid, ceasing only once the digestion fluids had reached pH 2.

In order to establish whether the shear and resulting particle size reduction imposed by the digestion apparatus, or the pH profile and the resulting effect on the enzyme activity was the dominant variable in controlling the rate of gastric digestion, additional experiments using the Glass stomach were run with a modified protocol using only gels prepared at pH 7. In this experiment, the pH was initially set to pH 3, after which no adjustments were made during the digestion protocol, and the results are presented in Figure 4b for comparison with both the Infogest and dynamic Glass stomach results. At the 30-min time point, all three data sets of gels prepared at pH 7 showed no significant difference in filterable nitrogen between the gastric digestion methods (*p* < 0.01). However, at the 60, 90 and 120 min points, the Glass stomach method utilising the dynamically controlled pH showed significantly greater amounts of filterable nitrogen from 60 min onwards than the two methods where the pH was pre-set to pH 3 (*p* < 0.01). The Glass stomach and Infogest methods with the pH pre-set at pH 3 showed identical trends in measured filterable nitrogen, despite a small difference in absolute amounts at the 30-min time point, which remained unchanged at subsequent time points. This suggests that the type of shear/digestion apparatus had no significant effect on the rate of digestion, at least with gels prepared at pH 7, and that the differences in gastric digestion kinetics was governed by the pH profile. Similar differences were found in a comparison between dynamic and pre-set pH gastric digestion systems by Mat et al. [52].

It was also noted that the pH of the digestion fluids in a number of other Infogest digestion experiments conducted in the authors laboratory and utilising a pre-set pH of 3, gradually shifted towards the pH at which the gel was prepared (data not shown). For this reason, the authors have avoided use of ‘static pH’ to describe the Infogest system. Gels prepared at pH 3 would have had no effect on the overall pH of the digestion fluids. However, gels prepared at pH 5 and 7 gradually shifted the pH higher during digestion in the Infogest system. In the Glass stomach, the pH of the digestive fluids increased substantially immediately following addition of the minced gels prepared at pH 5 and 7, and gradually reduced as HCl was added. These pH changes would have had significant impacts on the activity of Pepsin, as discussed earlier.

The results of these experiments suggest that a dynamic model may provide more bio-relevant experimental conditions of gastric digestion than the Infogest method, particularly with more solid-like/gelled foods having higher pH values. In these types of food systems, there is a strong likelihood that the pH of the food, as well as buffering action of the proteins, will over the course of digestion raise the pH above the pre-set pH of 3. This is particularly important when studying rates of gastric digestion, as higher pH values dramatically affect the activity of Pepsin. Indeed, the Infogest protocol itself indicates that it is not suitable for evaluation of digestion kinetics [53].

The gastric OPA results for gels prepared at pH 3 showed that measured OPA reactivity was linked to filterable nitrogen content and that protein hydrolysis was relatively complete after 30 min. It is therefore likely that the majority of the differences between the Infogest and Glass stomach data for gels prepared at pH 3 could be attributed to the differences in the shear imposed by the digestion systems, with the Infogest approach only being able to fully break down the sticky aggregates after 120 min. This is not to say that Pepsin activity had no impact, as the pH in the Glass stomach would have lowered towards pH 2, leading to high Pepsin activity, while that in the Infogest system would have remained at pH 3. More so, that no significant difference was observed. The relatively small sizes of the particles of gels produced at pH 3 would have yielded a high surface area, and this in turn would have allowed for rapid enzymatic digestion in instances when enzyme activity was high. During digestion of gels prepared at pH 3, in both the Infogest and Glass stomach methods, the pH of the digestion fluids would have been at or below pH 3 at all times, leading to relatively high Pepsin activity (compared to higher pH values) and combined with the high surface area of the gel particles, would have led to rapid hydrolysis of the protein.

The Infogest and pre-set Glass stomach procedures, having a pre-set pH of 3, yielded a Pepsin activity no higher than 40% of its maximal activity at pH 2, and likely even lower during digestion of gels prepared at pH 5 and 7. Initially, the pH of the dynamic Glass stomach digestion fluids would also have shifted higher than pH 3. However, as digestion progressed and the addition of HCl reduced the pH, the activity of Pepsin would have increased substantially and helped to break down the gel. This led to an increase in filterable nitrogen in the dynamic Glass stomach compared to the digestions with a pre-set pH of 3 and explains the results of the gels prepared at pH 7 in Figure 4b. While it is clear from these results that the digestion method plays a part in the level of filterable nitrogen, it is far exceeded by the effect of the aggregate sizes entering into the gastric phase, as evidenced by the much lower filterable nitrogen levels for the gels prepared at pH 7 than those prepared at pH 3 and 5.

The trends for the gels prepared at pH 5 were also a consequence of the initially high pH, which was sub-optimal for Pepsin in both digestive systems, although the Pepsin activity would have gradually improved as the pH decreased. Coupled with the small particle size and hence, high surface area of the minced gels prepared at pH 5, the Infogest method yielded lower OPA readings than the dynamic Glass stomach, as shown in Figure 5a,b respectively.

The results of the gastric digestion experiments show that the rate of digestion is primarily controlled by the size of the particles entering into the gastric phase, which is a function of the elasticity of the gels and the shear the gels are exposed to during the oral processing step. However, the pH also impacts the rate of digestion via its impact on Pepsin activity. A minor effect was also seen in the results for the digestion of gels prepared at pH 3 and has been attributed to the apparatus used in the experiments.

#### 3.2.3. Simulated Intestinal Digestion

In this study, gastric digestion aliquots collected after 30, 60, 90 and 120 min were exposed to the simulated intestinal digestion protocol and analysed for OPA reactivity. The raw OPA data was found to be strongly correlated to the level of filterable nitrogen measured for each collection time point, that is, the amount of solubilised protein that would enter the intestine as gastric digestion proceeds. Hence, the OPA results were normalised to the amount of filterable nitrogen, leading to negligible differences within the data set of each gel pH and digestion method. Hence, these were averaged to yield a single curve and are presented in Figure 6a,b. This suggests that the time point at which the solubilised protein exits the stomach does not influence intestinal digestion rates in vitro.

The normalised OPA levels at 0 min were in the order pH 7 > pH 3 > pH 5 for the pH values at which the gels were prepared, and the trend of these differences remained throughout the 2 hr sampling window from digesta originating from both the Infogest and Glass stomach gastric digestions. Given that the difference in OPA reactivity was evident from the 0-min time point onwards, it can be attributed to differences arising from gastric digestion, and may be a consequence of the differing nitrogen values in the gastric filtrates, which were not fully off-set by the extent of protein hydrolysis. For example, gels prepared at pH 7 yielded much lower filterable nitrogen values than those prepared at pH 3 and pH 5. Hydrolysis of the gels prepared at pH 7 would have been hindered by the size of the minced particles, leading to low OPA values. However, the excluded volume created by the gel particles may have led to a higher effective Pepsin concentration, meaning that raw OPA values were not reduced to the same extent as the level of filterable nitrogen. Hence, when dividing the raw OPA values by the filterable nitrogen values, this may have led to the relatively higher normalised OPA value for the gels prepared at pH 7 at the start of the intestinal digestion process. If the initial differences in normalised OPA values are taken into account, the differences in intestinal digestion over time as measured by subsequent OPA readings would be negligible for the digesta stemming from the Infogest gastric phase (Figure 6a). The digesta from the Infogest and Glass stomach of the gels prepared at pH 5 showed no significant difference between them during intestinal digestion (*p* < 0.01), which also supports this argument. Differences in intestinal digestion rates after 30 min were however observed between the Infogest and Glass stomach digesta for gels prepared at pH 3 and 7, with digesta from the Infogest method yielding higher normalised OPA values than those from the Glass stomach. This result may also have arisen from differences in the filterable nitrogen between the gastric digestion methods. In the intestinal phase however, the impact of excluded volume would have been lower than during the gastric phase, as the largest particles were removed from the system during the filtering step. Assuming this was the case, the higher filterable nitrogen values, as seen in the Glass stomach gastric phase, would have led to a divergence from the intestinal data arising out of the Infogest gastric phase. While these differences are significant (*p* < 0.01), it can be said that the differences in the intestinal digestion of different WPI protein structures are relatively small in comparison to those observed during gastric digestion. The findings in this work support those of a number of other authors in that the rate of overall digestion is strongly influenced by digestion in the gastric phase and, in an in vivo setting, would be linked to gastric emptying [23,48,54]. The rate of intestinal digestion was far less affected by structure and could likely be explained by intrinsic variations arising from concentration differences created by the gastric digestion step. Once these factors were taken into account, no significant differences could be definitively identified in the intestinal digestion phase and linked with gel microstructure.

## 4. Conclusions

Protein gels with different structural and mechanical properties can be produced via heat set gelation by adjusting the pH prior to heating. Mechanical chewing for the oral phase was simulated by mincing each gel, which produced particles and aggregates of varying size, which were determined by the initial material properties of the gels. The extent of digestion as measured by filterable nitrogen through the simulated gastric phase was primarily related to differences in the size of the particles entering the gastric phase. A relationship was also noted between the activity of Pepsin in the gastric phase and the extent of digestion as measured by filterable nitrogen. This was the basis for the differences in the extent of digestion, as measured by both filterable nitrogen and OPA between the Infogest and dynamic Glass stomach. No significant difference was found between the Infogest and pre-set pH Glass stomach models. No significant differences were found in the rate of enzymatic digestion in the intestinal phase for digesta of gels prepared at different pH values and passed through the dynamic Glass stomach. Small differences in intestinal digestion were seen for samples having been processed through the Infogest gastric method, but these differences may be due to differences in the ratio of intestinal enzymes to available protein.

Overall, the key finding of this work is that the rate of in vitro simulated digestion of WPI gels is strongly governed by the mechanical properties of the gel and that oral processing has a very significant role as a rate determining step. The firmness and elasticity of the gel relate to its break-down behaviour and affect the size of gel particles produced following simulated oral processing. It was found that gastric digestion had a relatively small influence on the gel particle sizes over the two-hour period studied. It was also found that the rate of digestion was also (albeit to a lesser extent than the gel mechanical properties) affected by the pH of the substrate and digestion fluids. Under the shear conditions investigated, the particle sizes of the substrates at the beginning of gastric digestion (i.e., following oral processing) were found to be governed by the mechanical properties of the protein network, which in turn translated to reduced digestion over the simulated gastric protocol. This work highlights the differences in the rate of gastric digestion arising from oral processing, which can be attributed to the material properties of the substrate.

This work highlights the potential for the material properties of a food to be tailored to achieve faster or slower digestion rates. In turn, it is possible that controlling the rate of digestion may have biological consequences: e.g., to satiety, gut microbiota and others. However, further research would be required to facilitate those conclusions. In particular, focus could be directed to investigating the bioaccessibility/bioavailability of digesta from gels produced with marked differences in initial structural and mechanical properties.

## Figures and Tables

**Figure 1 foods-10-01066-f001:**
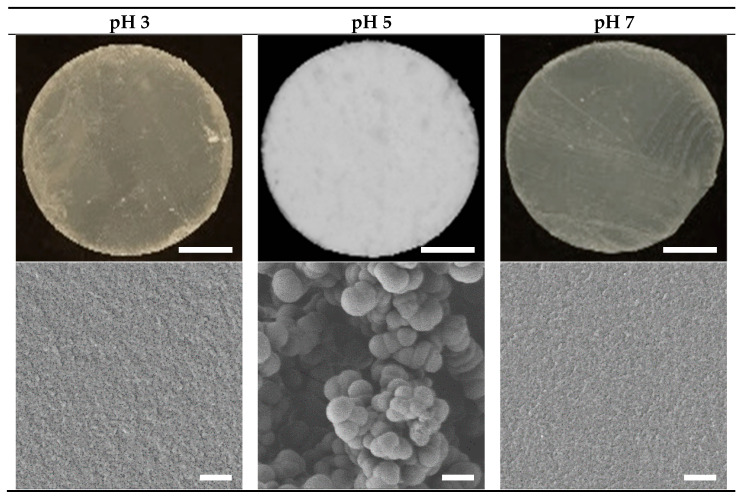
Photographs (top) and scanning electron micrographs (SEM, bottom) of the 15% WPI gels prepared at pH 3, 5 and 7. Scale bars represent 5 mm for the photographs and 2 μm for the SEM images, respectively.

**Figure 2 foods-10-01066-f002:**
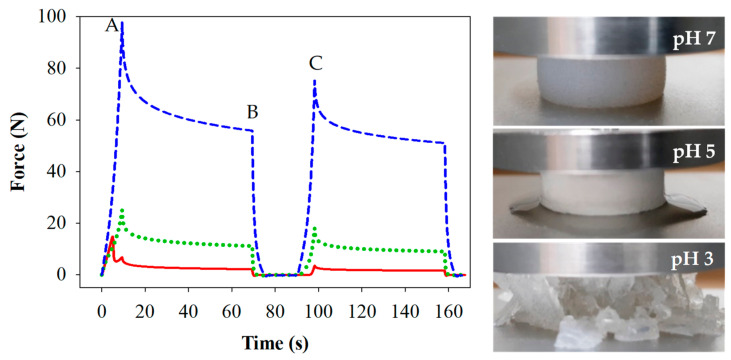
Force/time curves and images from compression/relaxation tests of 15% WPI gels prepared at pH 3 (

), 5 (
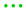
) and 7 (
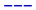
). Measures of firmness (A, B and C) are indicated for samples prepared at pH 7. Images were captures at point A for each of the gels.

**Figure 3 foods-10-01066-f003:**
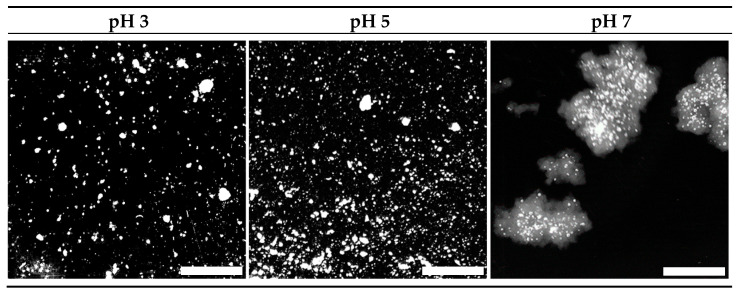
Macro photographs of 15% WPI minced gels (to simulate chewing) prepared at pH 3, 5 and 7. Scale bar represents 5 mm. Images have been converted to black and white for enhanced clarity.

**Figure 4 foods-10-01066-f004:**
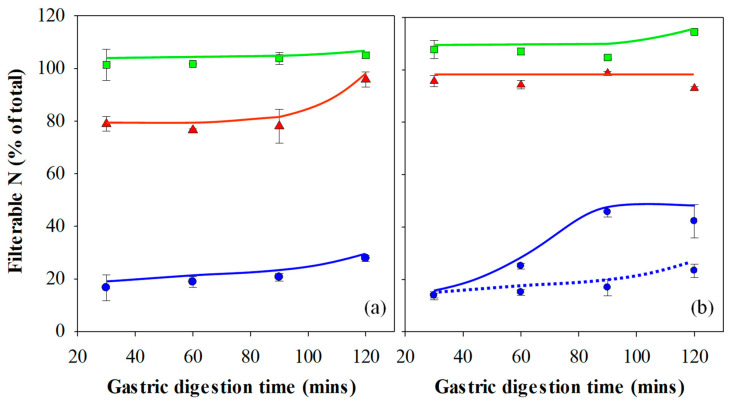
Filterable nitrogen following in vitro gastric digestion of 15% WPI gels prepared at pH 3 (▲), 5 (◼) and 7 (◾) using the (**a**) Infogest and (**b**) Glass Stomach digestion protocol. Gels prepared at pH 7 and digested using the static Glass stomach protocol with a pre-set pH of 3 is shown for comparison (**‧‧‧‧**). Results are presented as the mean ± standard error.

**Figure 5 foods-10-01066-f005:**
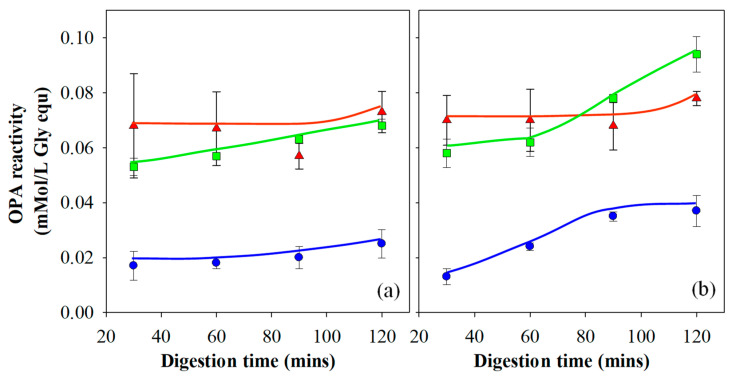
OPA reactivity following in vitro gastric digestion of 15% WPI gels prepared at pH 3 (▲), 5 (◼) and 7 (◾) using the (**a**) Infogest and (**b**) Glass Stomach digestion protocol. Results are presented as the mean ± standard error.

**Figure 6 foods-10-01066-f006:**
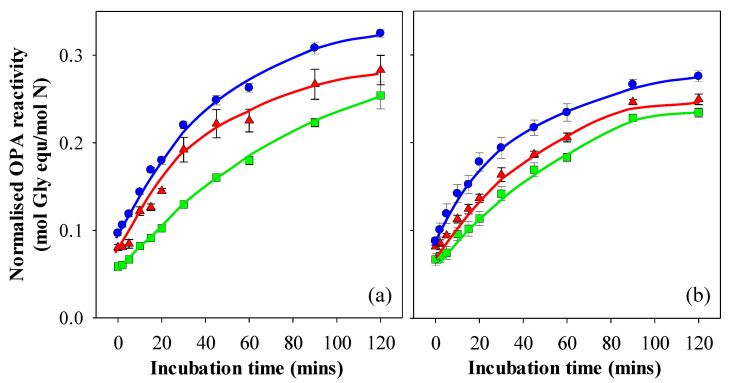
Normalised OPA reactivity of intestinal digesta following in vitro gastric digestion of 15% WPI gels prepared at pH 3 (▲), 5 (◼) and 7 (◾) using the (**a**) Infogest and (**b**) Glass Stomach digestion protocol. Results are presented as the mean ± standard error of samples stemming from the 30, 60, 90 and 120 min gastric digestions.

**Table 1 foods-10-01066-t001:** Average results from the large deformation rheology analysis for compression and relaxation of 15% WPI gels prepared at pH 3, 5 and 7. Results are presented as the mean ± standard error. A statistically significant difference between the different gels in each row is indicated by a small letter (*p* < 0.05).

Output Variable	pH 3	pH 5	pH 7
Load at point A = first peak (N)	18.8 ± 2.8 ^a^	26.6 ± 1.6 ^a^	101.9 ± 5.8 ^b^
Load at point B = end of hold (N)	2.7 ± 0.3 ^a^	11.9 ± 0.7 ^b^	56.3 ± 2.5 ^c^
Load at point C = second peak (N)	5.0 ± 0.6 ^a^	19.7 ± 1.2 ^b^	78.9 ± 4.4 ^c^
‘Lost firmness’ = LF = (A-B)/A (%)	84.3 ± 0.9 ^c^	55.2 ± 0.3 ^b^	44.4 ± 0.8 ^a^
‘Recovered firmness’ = RF = C/A (%)	27.9 ± 1.5 ^a^	73.9 ± 0.3 ^b^	77.4 ± 0.2 ^c^

## Data Availability

The data that support the findings of this study are available on request from the corresponding author.

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
