# Peer review of "WPI Gel Microstructure and Mechanical Behaviour and Their Influence on the Rate of In Vitro Digestion"

_foods, 2021, doi:10.3390/foods10051066_

Round 1

Reviewer 1 Report

The article « WPI gel microstructure and mechanical behaviour, and their influence on the rate of in-vitro digestion» presents interesting results dealing with the differences in the rate of gastric digestion, arising from oral processing, which can be attributed to the material properties of the substrate. However, several points need to be explained or precise to upgrade the quality of the manuscript:

  • P1, line 26: Could the authors replace the second use of the word comprise?
  • P2, line 78: Could the author precise that the Infogest Protocol is a harmonised method, used in a large number of laboratory because it could be important in the discussion and to argue the choice of this method.
  • P5, line 187: Is this method simulating the chewing already published, and in accordance with results obtained in vivo? I think that a reference is needed to justify the use of this protocol even if a part of the answer could be find page 10, line 369.
  • Figure 2: The green points are difficult to see on this representation.
  • P9, line 338: Could the authors precise the unit of the mean diameter values? How or was the homogeneity of the samples verify?
  • P11, line 435: I draw the attention of the authors to the fact that they do not precise if the homogeneity of the samples prepared was checked during the experimentation. Could the larger protein particles at pH 3 due to a problem of inhomogeneity at this pH? Do the authors think this could be specified or clarify in the discussion?
  • P11, line 447: Could the authors precise if these two methods were already compared for this kind of matrix and how the most important difference could explain the small differences in terms of final result? What about the importance of the kinetic aspects for the nutrients absorption?
  • P12, line 469: Could the authors precise here how the pH of sample preparation could impact the hydrolysis process by pepsin, because the pH in the gastric compartment is always stabilized, in both model, to permit the enzyme activation. It refers also to the remark line 511 (page 13), where it is precise that the pH is un-monitored in the Infogest protocol, but the gastric effluent plays a role in the stabilization of the ph. Do the authors know the differences observed in the pH variation, the pH profile from line 532, between both methods?

This article is very interesting because it highlights an important issue in the use of static or dynamic models to simulate in vitro GI digestion of food with different structure related to the pH of preparation. The final point, at intestinal level, do not seems to be impacted, but the kinetic aspects are of great interest in terms of digestion understanding and health benefit.  

Author Response

R1:

Comments and Suggestions for Authors

The article « WPI gel microstructure and mechanical behaviour, and their influence on the rate of in-vitro digestion» presents interesting results dealing with the differences in the rate of gastric digestion, arising from oral processing, which can be attributed to the material properties of the substrate. However, several points need to be explained or precise to upgrade the quality of the manuscript:

P1, line 26: Could the authors replace the second use of the word comprise?

AU response: The work ‘comprising’ has been changed for ‘consisting’, to read: 

“….which usually comprises more than 50 % of the total protein, with the remainder comprising consisting of the other serum proteins….”

P2, line 78: Could the author precise that the Infogest Protocol is a harmonised method, used in a large number of laboratory because it could be important in the discussion and to argue the choice of this method.

AU response: The text has been modified as follows:

“An international collaboration created the Infogest protocol, which proposed a harmonized method for digesting foods using an in-vitro system [35]. This was superseded by Infogest 2.0 [36], and has been applied for the in-vitro study of digestion in a large number of laboratories throughout the world.”

P5, line 187: Is this method simulating the chewing already published, and in accordance with results obtained in vivo? I think that a reference is needed to justify the use of this protocol even if a part of the answer could be find page 10, line 369.

AU Response: The method used to simulate the chewing process in this work is based on the recommendations by Minekus et al., (2014) within the Infogest protocol. This has now been referenced in the text.

Figure 2: The green points are difficult to see on this representation.

AU Response: The green points have been darkened, and the arrows removed providing greater clarity to the Figure.  

P9, line 338: Could the authors precise the unit of the mean diameter values? How or was the homogeneity of the samples verify?

AU Response: the units for measure (mm) have been added, as follows:

“Image analysis yielded mean diameter values of 1.32 ± 0.16, 1.29 ± 0.12 and 4.62 ± 0.75 mm for the minced gels prepared at pH 3, 5 and 7 respectively.”

The method used to capture and analyse the particle size data has been modified for additional clarity, as follows:

“The photographs were uploaded to Fiji (Image J) v. 1.52p [38]. The images were converted to black and white and their contrast and brightness levels were adjusted to aid in visualising the particles present. Image analysis was conducted by following the protocol in the Image J User Guide v. 1.46r [39]. An approximate value of the mean diameter of the particles was calculated from the average particle area by assuming perfect sphericity.”

  • Reference numbers have been updated accordingly to account for the newly introduced reference.

In addition, the homogeneity of the gels themselves can be observed in the cross-sectional images shown in Figure 1. Observations showed that gel formation began almost instantly upon putting the sausages in the water bath and cooking was complete within around 5 mins. To provide a safety margin, 10 mins was used to fully ensure cooking was complete and the samples were homogeneous. This has been reflected in the text as follows:

“Preliminary experiments in our laboratory (data not shown) demonstrates that 10 mins was sufficient to ensure the formation of homogeneous gels.”

P11, line 435: I draw the attention of the authors to the fact that they do not precise if the homogeneity of the samples prepared was checked during the experimentation. Could the larger protein particles at pH 3 due to a problem of inhomogeneity at this pH? Do the authors think this could be specified or clarify in the discussion?

AU Response: The simulated chewing step did result in a range of different sized particles for the pH 3 gels, as suggested by the reviewer. The image analysis (described in section 2.2.4.6: “Analysis of digested material”) allowed us to visualise this range, and to note the similarity in range between the smaller and larger particles formed from the pH 3 and pH 5 gels. It is therefore our hypothesis that differences in sieving were observed due to the stickiness of the pH 3 gels forming effectively larger aggregated particles. Additional information to describe the visualisation method has been added, as described in the response to the previous comment (above).

P11, line 447: Could the authors precise if these two methods were already compared for this kind of matrix and how the most important difference could explain the small differences in terms of final result? What about the importance of the kinetic aspects for the nutrients absorption?

AU Response: It is our belief from extensive reviews of the literature that comparison between the Glass stomach and Infogest methods on WPI gels has not been investigated in the same manner as has been performed in our experiments. In this work, the difference in mechanical properties of the gel was attributed to the breakdown behaviour of the gel which in turn impacted the rate of digestion in the in vitro stomach models. This has been discussed throughout the paper but most notably in section 3.2.2. and the conclusions (lines 682-686). The authors have deliberately not ventured into discussions around the kinetic absorption of nutrients as this sits outside the remit of this investigation and would be an extensive body of work in itself.

P12, line 469: Could the authors precise here how the pH of sample preparation could impact the hydrolysis process by pepsin, because the pH in the gastric compartment is always stabilized, in both model, to permit the enzyme activation. It refers also to the remark line 511 (page 13), where it is precise that the pH is un-monitored in the Infogest protocol, but the gastric effluent plays a role in the stabilization of the ph. Do the authors know the differences observed in the pH variation, the pH profile from line 532, between both methods?

AU Response: The pH of the gel sample can shift the initial pH of the gastric vessel if higher than that of the simulated gastric fluid (i.e. the gels prepared at pH 5 and pH 7). Here, the intorduction of the gel will shift the pH of the ‘adjusted simulated gastric fluid + simulated salivary fluid + gel system’ to a less acidic pH, as equilibrium is reached between the aqueous phase within each gel particle and the external vessel pH. This is discussed in lines (554-563). The authors observed these pH shifts after the pH was initially adjusted to pH 3 and equilibrium was reach between the sample and the simulated fluids. These shifts have been quantified in a subsequent study yet to be publish, discussed in lines 554-556. However, our observations for the current study are qualitative.

Reviewer 2 Report

The paper was writen in a clear and precise way. The material and methods does provide all the necessary information. The results are discussed with a great care of details.

However, I would like you to add/clarify some information:

Lane 126: What was the volume that was pured to the tubes?

Lane 129: Did you measure the temperature inside the sausage? Was 10 min. enough for the temperature gradient to become equal in the whole sample?

Lane 140: Could you please explain this? How did you measure the strength of the sample to decide which load cell use?

Lane 211: How long did you freeze the samples? Was it necessary to freeze the sample? 

Lanes 234-236: Simply write what you have done with the sample. Referring to Infogest procedure, it might be not clear enough to which part of bullet 2.2.4.2. you refer to. 

Figure 2: Which stage of the compression/relaxation test does the pictures represent?

Table 1: The a, b, c value standing for significant difference should be placed after the value not after SD value. 

Lane 338: what are the units of diameter? mm?

Lane 451: delete 'and'

Author Response

This article is very interesting because it highlights an important issue in the use of static or dynamic models to simulate in vitro GI digestion of food with different structure related to the pH of preparation. The final point, at intestinal level, do not seems to be impacted, but the kinetic aspects are of great interest in terms of digestion understanding and health benefit. 

R2:

The paper was writen in a clear and precise way. The material and methods does provide all the necessary information. The results are discussed with a great care of details.

However, I would like you to add/clarify some information:

Lane 126: What was the volume that was pured to the tubes?

AU Response: The text has been amended to include the approximate length of the sausages and the volume as follows:

“…and then poured into flexible plastic sausage-like tubes (~20 mm in diameter, ~400-500 mm in length, equating to approximately 150 mL volume).”

Lane 129: Did you measure the temperature inside the sausage? Was 10 min. enough for the temperature gradient to become equal in the whole sample?

AU Response:  The cooking protocol/time applied to this study was based on preliminary experiments in our laboratory that demonstrated 10 min was sufficient to ensure the formation of homogeneous gels. This has been noted in the text as follows:

“The sausages were cooked for 10 mins before being removed and placed horizontally in a tray and stored for approximately 12 hr at 4 °C. Preliminary experiments in our laboratory (data not shown) demonstrates that 10 mins was sufficient to ensure the formation of homogeneous gels.”

Lane 140: Could you please explain this? How did you measure the strength of the sample to decide which load cell use?

AU Response: Load cells are selected to ensure that it’s limits would not be exceeded during the tests, based on preliminary analysis. This is now reflected in the text as follows:

“The WPI cylinders were individually loaded onto a universal testing machine (Instron 5564, Massachusetts, USA) fitted with either a 100 or 500 N load cell, depending on the strength of the sample being tested, determined through preliminary analysis (data not shown) to ensure the limits of each load cell were not exceeded.”

Lane 211: How long did you freeze the samples? Was it necessary to freeze the sample?

AU Response: While gastric digesta can run immediately into an intestinal in-vitro model, it was necessary for the process in our laboratory that samples be collected and frozen prior to the intestinal step. The following text has been included in lines 254-255: “On average, samples were stored for ~30 days. No one sample was stored for longer than 60 days.”

Lanes 234-236: Simply write what you have done with the sample. Referring to Infogest procedure, it might be not clear enough to which part of bullet 2.2.4.2. you refer to.

AU Response: we thank the reviewer for this response. To avoid repetition, we have outlined the sample collection protocol applied to both the Infogest and Glass stomach methods as a separate section, as follows:

 “2.2.4.4. Gastric phase sample collection (Infogest and Glass stomach methods)

At each time point, ‘undigested’ material was separated from ‘digested’ material by filtration through a sieve with an approximate mesh size of 1.5 mm, and the retentate washed with a small amount of deionised water. The filtrate and washings were weighed, and an aliquot taken to determine the nitrogen content of the filtrate.

A sample of each gastric filtrate (20 g) was mixed with intestinal buffer (11 mL, 1.25× final concentration pH > 9) raising the pH > 8 to cease enzyme activity and immediately frozen for storage prior to subsequent intestinal digestion.”

Figure 2: Which stage of the compression/relaxation test does the pictures represent?

AU Response: The images were taken at point A as was indicated by the arrows. For clarity, we have removed the arrows and instead added this information into the Figure caption.

Table 1: The a, b, c value standing for significant difference should be placed after the value not after SD value.

AU Response: having checked through previous papers published in Foods we have found that others have also reported the statistical value for significance after the SD value, and we respectfully ask that we may keep our formatting as written.

e.g.

Foods 2021, 10(5), 928; https://doi.org/10.3390/foods10050928

Foods 2021, 10(4), 843; https://doi.org/10.3390/foods10040843

Lane 338: what are the units of diameter? mm?

AU Response: the units for measure (mm) have been added, as follows:

“Image analysis yielded mean diameter values of 1.32 ± 0.16, 1.29 ± 0.12 and 4.62 ± 0.75 mm for the minced gels prepared at pH 3, 5 and 7 respectively.”

Lane 451: delete 'and'

AU response: the word ‘and’ has been deleted. We thank the reviewer for noticing this error.